# Analysis of the Effects of Pentose Phosphate Pathway Inhibition on the Generation of Reactive Oxygen Species and Epileptiform Activity in Hippocampal Slices

**DOI:** 10.3390/ijms25031934

**Published:** 2024-02-05

**Authors:** Daria Ponomareva, Anton Ivanov, Piotr Bregestovski

**Affiliations:** 1Department of Physiology, Kazan State Medical University, 420012 Kazan, Russia; darya.ponomareva@kazangmu.ru; 2Institute of Neuroscience, Kazan State Medical University, 420012 Kazan, Russia; 3INSERM, Institut de Neurosciences des Systèmes (INS), UMR1106, Aix-Marseille Université, 13005 Marseille, France; anton.ivanov@univ-amu.fr

**Keywords:** pentose phosphate pathway, glucose-6-phosphate dehydrogenase, glucose metabolism, reactive oxygen species, H_2_O_2_ release, epilepsy, brain slices, fluorescent analysis, electrophysiology

## Abstract

The pentose phosphate pathway (PPP) is one of three major pathways involved in glucose metabolism, which is regulated by glucose-6-phosphate dehydrogenase (G6PD) controls NADPH formation. NADPH, in turn, regulates the balance of oxidative stress and reactive oxygen species (ROS) levels. G6PD dysfunction, affecting the PPP, is implicated in neurological disorders, including epilepsy. However, PPP’s role in epileptogenesis and ROS production during epileptic activity remains unclear. To clarify these points, we conducted electrophysiological and imaging analyses on mouse hippocampal brain slices. Using the specific G6PD inhibitor G6PDi−1, we assessed its effects on mouse hippocampal slices, examining intracellular ROS, glucose/oxygen consumption, the NAD(P)H level and ROS production during synaptic stimulation and in the 4AP epilepsy model. G6PDi−1 increased basal intracellular ROS levels and reduced synaptically induced glucose consumption but had no impact on baselevel of NAD(P)H and ROS production from synaptic stimulation. In the 4AP model, G6PDi−1 did not significantly alter spontaneous seizure frequency or H_2_O_2_ release amplitude but increased the frequency and peak amplitude of interictal events. These findings suggest that short-term PPP inhibition has a minimal impact on synaptic circuit activity.

## 1. Introduction

The mammalian brain is an organ that consumes a large amount of energy. Its activity largely depends on glucose being delivered by blood into the brain and a tight regulation of glucose metabolism is critical for reliable brain functioning [1,2]. Glucose is converted to glucose-6-phosphate (Glu6P) under the action of hexokinase, which primarily controls the metabolism of this monosaccharide [3,4]. Glu6P is directed by various enzymes along three main pathways: glycolysis, glycogen synthesis and the pentose phosphate pathway (PPP) (Figure 1). Glycolysis is used for energy production, while the PPP facilitates the production of NADPH, a reduced form of nicotinamide adenine dinucleotide phosphate (NADP). NADPH allows to preserve the reduced form of glutathione, the main cellular antioxidant that suppresses reactive oxygen species (ROS) and thus balances oxidative stress [5].

Glucose catabolism in the PPP is channeled by the enzyme Glucose-6-phosphate dehydrogenase (G6PD), which converts Glu6P to 6-phosphoglucono-δ-lactone with parallel reduction of NADP. G6PD has been intensively studied in mammalian cells, since a partial deficiency in this enzyme underlies the most common form of nonimmune hemolytic anemia [6] and a full depletion of G6PD in mammals leads to embryonic lethality [7]. It has been suggested that G6PD is involved in the pathology of various human diseases such as heart failure [8], hypertension and cancer [9]. In addition, it has been indicated that G6PD deficiency may be a predisposing factor for rhabdomyolysis following a tonic–clonic seizure [10] and progressive myoclonic epilepsy can be associated with a mutation in G6PD [11]. G6PD dysfunction leads to a decrease in the concentration of NADPH and the reduced form of glutathione, which, in turn, causes increased oxidative stress [12,13]. It has been also reported that patients with Alzheimer’s disease have increased expression of G6PD in the human hippocampus, indicating that the activity of the enzyme may be particularly important in this part of the brain [14,15].

The hippocampus is involved in in both cognition and regulating emotions. The dorsal hippocampus is primarily involved in cognitive functions, such as learning and memory related to navigation, exploration and locomotion. In contrast, the ventral hippocampus is linked to motivational and emotional behavior [16,17]. These functions are supported by the unique anatomical, morphological, molecular and electrophysiological features of hippocampal cells [18].

Previous work in hippocampal slices also showed that the rapid release of hydrogen peroxide is mainly supplied by NADPH oxidase (NOX), which is a trigger for epileptic seizures [19]. In this pathological condition, NADPH can be used by NOX for ROS production [5]. Active NOX generates superoxide by transferring an electron from NADPH in the cytosol to oxygen in the extracellular space [20]. Thus, NADPH, which is also produced in the PPP, could potentially be involved in two competing reactions: NOX, which produces ROS, and the PPP, which neutralizes them.

Although several pathological disorders associated with G6PD deficiency are well documented, the consequences of PPP modulation due to G6PD inhibition require in-depth analysis. In particular, it is important to elucidate how the suppression of G6PD function modulates the baseline level of ROS and changes in ROS caused by synaptic activation, as well as the effect of G6PD inhibition on seizure generation in models of epilepsy.

To clarify some of these issues, we have selected a recently proposed effective inhibitor of G6PD, a nonsteroidal small molecule (G6PDi−1) [21]. The G6PDi−1 acts by blocking or reducing the activity of the G6PD enzyme, which leads to the inhibition of the pentose phosphate pathway. In our study, G6PDi−1 was used as a tool to analyze the following questions: (i) how inhibition of the G6PD enzyme affects glucose and oxygen consumption during synaptic stimulation of neurons in brain slices; (ii) how inhibition of G6PD by G6PDi−1 affects the base level of NAD(P)H and ROS and their changes during synaptic stimulation; and (iii) how G6PDi−1 acts on ROS production during seizure-like activity in the 4-aminopyridine (4AP) model of epilepsy in hippocampal slices. For this purpose, we simultaneously measured electrical (local field potentials, LFPs) and metabolic network parameters (extracellular oxygen, glucose, H_2_O_2_) in hippocampal slices.

## 2. Results

### 2.1. Effect of G6PDi−1 on Consumption of Glucose and Oxygen Induced by Shaffer Collaterals Stimulation in Hippocampal Slices

The PPP is an alternative glycolytic pathway for glucose metabolism [22]. To assess the extent of glucose metabolism modulation by PPP inhibition, we measured glucose and oxygen consumption induced by the synaptic stimulation of Shaffer collaterals in hippocampal slices when the PPP was suppressed ed by G6PDi−1, an inhibitor of the G6PD enzyme (Figure 1).

The changes in glucose and oxygen were measured simultaneously by glucose enzymatic microelectrodes and a Clark oxygen microelectrode in conjunction with an LFP registration. A scheme of the electrode’s position in the slice is shown in Figure 2a. The analysis, carried out on seven acute hippocampal slices, showed that, immediately after the installation of the glucose sensor, the baseline glucose level in different slices varied from 1.76 to 2.74 mM. The 30 s stimulation (200 ms pulses at 10 Hz) of Shaffer collaterals caused a remarkable consumption of glucose, ranging from 0.1 to 0.94 mM in different slices.

The following concentrations of G6PDi−1 were tested to determine glucose consumption: 50 nM, 500 nM, 2.5 μM, 5 μM. As the concentration increased, glucose uptake due to synaptic stimulation progressively decreased. The 2.5 μM concentration was found to be saturating and was used in the following experiments on hippocampal slices.

The traces shown in Figure 2b illustrate the glucose changes under the control conditions when adding 2.5 μM G6PDi−1 to the perfusion solution ACSF and after washing. The stimulation under control conditions (black) led to 0.62 mM consumption of glucose; during the application of G6PDi−1 (orange), the consumption of glucose decreased to 0.5 mM, and after washing (gray), the amplitude of consumption was elevated to 0.58 mM (Figure 2b,c).

The application of G6PDi−1, 2.5 μM did not lead to a significantly synaptically induced change in the rate of glucose consumption and its recovery kinetics. To evaluate the glucose consumption more accurately, we analyzed the area which was integral to changes in the glucose level in the slice from the moment of the onset of synaptic stimulation (30 s) to the moment when the glucose level was completely recovered (250 s). The relative glucose consumption at each stimulation was calculated relative to the last stimulation in the control.

G6PDi−1 caused a slowly developing decrease in synaptically induced glucose consumption. After 8–10 min of the inhibitor addition, glucose consumption decreased by 9.4 ± 4.5%, while its presence for 15–20 min resulted in a decrease by about 20%. As illustrated in Figure 2d, the average decrease was to 79.8 ± 4.0% compared to the control. Washing was accompanied by an increase in glucose consumption to 92.9 ± 7.3% (Figure 2d, *p* < 0.05, n = 6), which indicates the reversibility of the inhibitory effect of the compound.

To clarify whether G6PDi−1 affects the properties of synaptic transmission, we analyzed the electrophysiological characteristics of neuronal population activity following the application of G6PDi−1. Electrical signals (the presynaptic volley, postsynaptic response and spikes) were recorded using glass electrodes placed in the *stratum oriens* CA1 zone of the hippocampus (Figure 2a). During the G6PDi−1 application and after washing, the presynaptic volley of LFP component did not change compared to the control, indicating that a similar number of nerve fibers were activated (Figure 3a).

An example of a train of electrical signals during stimulation (30 s, 10 Hz) in the control, after the addition of G6PDi−1 and during washing with the insertion of single responses, illustrates the absence of any modulation (Figure 3b). To further validate these results, we performed an analysis of the LFP integrals, using the procedure previously described in detail [23]. The computer program separated each LFP, shifted the baseline to 0 and selected the region of integration. Population spikes were inverted and then the integral of the whole trace was calculated. On average, the mean integral of the population spikes (PS) of trains did not change; upon application of G6PDi−1 it became 99.3 ± 1.9% if compared with the control. The data summarized from six slices demonstrate that G6PDi−1 has no effect on the electrical activity on hippocampal slices (n = 6, Figure 3c). Furthermore, G6PDi−1 did not change the amplitude or kinetics of synaptically induced responses, indicating an absence of any effects on electrical activity.

These results strongly demonstrate that the decrease in glucose uptake induced by G6PDi−1 is not associated with a reduction in the efficiency of the synaptic stimulation of Shaffer collaterals.

We then analyzed the oxygen consumption induced by synaptic stimulation upon application of a G6PD inhibitor. The baseline oxygen level in the control was 16.3% and synaptic stimulation (10 Hz, 30 s) caused a remarkable decrease in its levels to 2.2% (Figure 4a). The decrease was reversible; after 200 s oxygen returned to its original level. The application of G6PDi−1 did not have a pronounced effect on the amplitude and kinetics of oxygen consumption. Figure 4b summaries relative changes in oxygen transients induced by synaptic stimulation in the control and during the G6PDi−1 application. In the presence of G6PDi−1 (2.5 µM), the average oxygen consumption integrals and oxygen transient amplitude were, respectively, 112.0 ± 4.5% (not significant, n = 7) and 106 ± 2.7% (not significant, n = 7) compared to the control.

Thus, inhibitor G6PDi−1 caused a small and reversible decrease in glucose consumption, whereas oxygen consumption tended to increase.

### 2.2. Effect G6PDi−1 on NAD(P)H/NAD(P) Levels

Reactions in the oxidative branch of the pentose phosphate pathway are considered to be one of the main producers of the reduced form of nicotinamide adenine dinucleotide phosphate (NADPH) [22]. NADPH and NADH have very similar optical properties, so NADPH is also expected to contribute to the overall autofluorescence signals to some extent. NAD(P)H autofluorescence was registered near the recording electrode in the stratum radiatum of the CA1 region of the hippocampus under the control conditions and within 20–30 min after the application of G6PDi−1.

Changes in the baseline level of NAD(P)H fluorescence under the influence of G6PDi−1 were evaluated relative to the level of NAD(P)H fluorescence under the control conditions. We observed a slight tendency towards an increase in NAD(P)H autofluorescence during PPP inhibition by 0.7 ± 0.2% compared to control (Figure 5a, n = 6). During synaptic stimulation, we observed a characteristic biphasic shape in the NAD(P)H signal under the control conditions: an initial short decline followed by a long-lasting overshoot. The same type of response was observed during the action of G6PDi−1 (2.5 µM) without significant changes in the dip and overshoot (Figure 5b).

### 2.3. Effect G6PDi−1 on Basic and Synaptically Induced Changes of Intracellular ROS Levels

The pentose phosphate pathway is a metabolic pathway in which glucose-6-phosphate is oxidized to generate pentose sugars, as well as reducing equivalents in the form of NADPH [22]. The PPP produces the reduced form of NADPH, which serves as an energy source for intracellular antioxidant systems; in particular, the glutathione and thioredoxin systems. We therefore analyzed intracellular ROS production upon the inhibition of the pentose phosphate pathway using the cell-permeable dye CellROX (see Section 4).

Fluorescence changes in a single slice under the G6PDi−1 application compared with the control are shown in Figure 6a and changes in six slices are shown in Figure 6b (n = 6, *p* < 0.05). The analysis showed that the inhibition of the PPP by G6PDi−1 causes an increase in the basal level of ROS in the cytoplasm; after the addition of G6PDi−1 (2.5 μM), fluorescence increased slowly and reached a maximum quasi-steady-state level after approximately 10 min. In the slice shown in Figure 5a, fluorescence increased by ≈0.3%. Across slices, the G6PDi−1-induced increase in the basal levels in hippocampal CA1 cells averaged 0.6 ± 0.17% (n = 6, Figure 6b).

To elucidate the effect of PPP inhibition on synaptically induced changes in ROS, we used the stimulation of Schaffer collaterals under control conditions and then in the presence of G6PDi−1. The increase in fluorescence was calculated as ΔF/F_0_, where ΔF is the change in fluorescence in the hippocampus and F_0_ is the fluorescence in the cortical area, since synaptic stimulation did not cause changes in fluorescence in this region.

Synaptic stimulation of Shaffer collaterals (10 Hz, 30 s) caused an increase in fluorescence (ΔF/F) under control conditions as well as in the presence G6PDi−1, reflecting an increase in intracellular ROS. The effect was reversible, and fluorescence returned to the initial level approximately 30–40 s after the end of the stimulation. (Figure 6c).

Importantly, the addition of 2.5 µM G6PDi−1 had no apparent effect on either the amplitude or kinetics of synaptically induced fluorescence changes (Figure 6c orange line). The change in fluorescence in response to stimulation was completely blocked in the presence of 1 µM TTX, indicating the synaptic nature of the change in fluorescence to stimulation.

In eight slices, the relative amplitude of fluorescence changes under the action of the G6PD inhibitor varied compared to the control, from 75.3% to 133.2%, with a mean value of 104.8 ± 6.2% (n = 8, Figure 6d), indicating the minor effect of G6PDi−1 on the synaptically induced increase in intracellular ROS.

### 2.4. Analysis of the G6PDi−1 Action on the Production of Hydrogen Peroxide and Seizure-Like Phenomena in the 4AP Model of Epilepsy

Previous studies demonstrated that spontaneous seizures were preceded by a rapid, high-amplitude release of hydrogen peroxide (H_2_O_2_) [19,24]. Notably, inhibiting NADPH oxidase, an enzyme responsible for the production of reactive oxygen species (ROS), has been shown to eliminate the rapid release of H_2_O_2_ and prevent the induction of seizures [19].

In this study, we sought to assess the effect of PPP inhibition, specifically the enzyme G6PD, on H_2_O_2_ production during epileptiform activity, as well as on the frequency of spontaneous seizures. For this purpose we applied a specific inhibitor of G6PD (G6PDi−1, 2.5 µM) on acute brain slices, generating SLEs induced by 4AP administration [25,26].

Application of 4AP (50 μM) resulted in hippocampal network hyperexcitability which manifested as interictal activity and seizure-like events (SLEs) (Figure 7a). The first SLEs became apparent 15–23 min (19.1 ± 1.5 min, n = 4) after the onset of perfusion in the 4AP-containing ACSF. The SLEs recorded in the *stratum oriens* of the CA1 hippocampal zone lasted between 40 and 78 s (60.7 ± 3.8 s, n = 15) and were characterized by a negative DC-shift in the local field potential (mean amplitude 2.8 ± 0.2 mV, n = 15). Simultaneous monitoring of local field potentials and extracellular H_2_O_2_ revealed that all spontaneous seizures were associated with an especially high and fast release of H_2_O_2_ both in the control (4AP alone) and following the G6PDi−1 application (Figure 7a).

Special attention is given to the analysis of spontaneous SLEs frequency. We observed an increase in seizure occurrence in two slices during both the application of G6PDi−1 (from three to four and from four to six SLEs per hour) and the washing phase (Figure 7b). In contrast, the other two slices exhibited a tendency toward a decrease (from six to five and from five to four SLEs per hour) in seizure frequency (Figure 7c). The difference in the effect direction may be due to the difference in the hippocampal region from which the brain slice came from. Indeed, when cutting the sagittal slices, we did no control whether they were from dorsal or ventral hippocampus (see Section 3).

We also monitored the production of H_2_O_2_ during spontaneous seizure-like events (SLEs), observing variations in peak amplitude ranging from 2 to 8 μM across different slices. The changes in the peak amplitude of H_2_O_2_ production during the application of G6PDi−1 and after washing, corresponding to the slices depicted in Figure 6b,c, are visually represented in graphs 6d and 6e, respectively. Importantly, there was a tendency towards a gradual reduction in the peak amplitude of hydrogen peroxide production upon the addition of G6PDi−1 and after the washing phase, as depicted in Figure 7d,e. This pattern was consistently observed in other slices as well. On average, the relative peak amplitude of H_2_O_2_ release during spontaneous seizure-like events (SLEs) was 94.9 ± 2.8% upon the application of G6PDi−1 (median difference—5.05% [CI—15.3%, 7.69%]), and during washing, it was 93.4 ± 4.9% (median difference—10.5% [CI—25.1%, 4.9%]) compared to the control in the 4AP model of epilepsy (Figure 7f). Thus, the PPP inhibition reduced the release of H_2_O_2;_ however, the size of the effect is too small to conclude its statistical significance.

We also analyzed the release of H_2_O_2_ associated with interictal-like events (IIEs). Interestingly, under the control conditions (4AP alone), the amplitudes of H_2_O_2_ transients generated during IIEs were significantly lower than in the case of SLEs (median difference—2.98μM [CI—3.58μM, −1.49 μM]), while during GP6H inhibition, these differences almost disappeared (median difference—0.891 μM [CI—1.92 μM, −0.22 μM]). Additionally, the frequency of IIEs accompanied by H_2_O_2_ release increased in the presence of the inhibitor (11 vs. 9 IIEs per hour); however, the amplitude of release decreased (median difference -0.205 μM [CI—1.85 μM, 0.589 μM]). These observations may suggest two opposing processes: the facilitation of H_2_O_2_ release due to the weakening glutathione-dependent antioxidant defense and a decrease in NOX-mediated H_2_O_2_ production. Indeed, both processes require NADPH, the production of which is ensured by the PPP.

## 3. Discussion

Glucose, the main energy source for biological tissues, is utilized via three main pathways: glycolysis, glycogen synthesis and the pentose phosphate pathway (PPP) (Figure 1). The PPP comprises two separate branches, the oxidative and non-oxidative [4]. The oxidative branch primarily depends on glucose 6-phosphate dehydrogenase (G6PD). This branch ensures the production of NADPH, which is required to maintain high the level of the reduced glutathione—the main cellular antioxidant. Therefore, the dysfunction of G6PD may result in the oxidative stress.

A number of studies indicate that oxidative stress and the production of ROS contribute to the pathogenesis of neurological diseases such as Alzheimer’s disease, Parkinson’s disease, Huntington’s disease, strokes, amyotrophic lateral sclerosis and the pathogenesis of epilepsy [4,27,28,29]. Although animal studies have demonstrated the involvement of oxidative stress in the occurrence of spontaneous seizures in chronic epilepsy [30,31,32], it is not yet clear whether excessive ROS production is a cause or a consequence (or possibly both) of seizure activity. Some studies indicate that oxidative stress is a consequence of seizures [24,33] while others consider it as the cause of epileptogenesis [19,34]. Although glucose metabolism via the PPP is crucial for antioxidant defense, it is still not clear if its impairment may contribute to the generation of epileptogenesis and/or seizures. 

Our study aims to clarify some of these questions. We present here an analysis of the effects of PPP inhibition on: (i) glucose and oxygen consumption, NAD(P)H level and ROS production during synaptic stimulation; (ii) seizure-like activity and ROS production in a 4AP model of epilepsy in hippocampal slices.

To inhibit the oxidative branch of the PPP, we used the recently proposed nonsteroidal molecule G6PDi−1, which has been described as an efficient and specific inhibitor of G6PD [21]. Previously, the steroid derivative dehydroepiandosterone (DHEA) has been used in many studies to suppress G6PD in in vitro and in vivo experimental models [35,36,37]. However, more recent observations demonstrated that DHEA is not a specific inhibitor of G6PD as it acts directly as a ligand for steroid, hormone and nuclear receptors, activating G-protein-coupled receptors and inhibiting voltage-gated T-type Ca^2+^ channels [38]. We therefore suggest that our approach using G6PDi−1 offers a more specific analysis of the consequences of PPP inhibition on synaptic function, network excitability, reactive oxygen species generation and epileptogenesis.

### 3.1. Effect of G6PDi−1 on Glucose Consumption Induced by Shaffer Collaterals Stimulation in Hippocampal Slices

Glucose and oxygen consumption are associated with neuronal network activity induced by the electrical stimulation of Schaffer collaterals in hippocampal slices. The addition of G6PDi−1 led to a gradual decrease in glucose consumption by about 20% compared to the control. The decrease in glucose uptake induced by G6PDi−1 is not associated with a reduction in the efficiency of the synaptic stimulation of Schaffer collaterals or synaptic function, since the inhibitor did not affect the strength of the local field potentials it evoked (LFPs). This contrasts with an earlier study which demonstrated that a widely used G6PD inhibitor, the steroid DHEA, caused a significant increase in LFP [39]. As mentioned above, DHEA is a nonspecific G6PD inhibitor. It may also cause the inhibition of the GABA_A_ receptors [40] or an increase in spontaneous glutamate release [41]. In contrast, G6PDi−1 exerts a more specific inhibition of G6PD [21].

A recent study showed that the inhibition of G6PDH activity does not affect the survival of wild-type motoneurons under cell culture conditions. In contrast, the specific inhibitor G6PDi−1 prevented neuronal death caused by the expression of mutated SOD1 or TDP43 proteins in a model of amyotrophic lateral sclerosis, indicating a beneficial effect of G6PDH activity on cells in pathological conditions [42]. Results from a study on cultured astrocytes demonstrated that G6PDi−1 (10 μM) inhibited 60% of the supply of electrons for a NQO1-catalysed β-lapachone-mediated reduction in the water-soluble tetrazolium salt 1, suggesting that NADPH, derived from PPP, is the main source of electrons for cytosolic NAD(P)H quinone oxidoreductase 1 (NQO1) [43]. Importantly, another study from this group provided compelling evidence that G6PDi−1 efficiently and specifically inhibits astrocytic G6PDH without affecting other oxidoreductases, indicating that G6PDi−1 is a suitable and specific tool for inhibiting G6PDH and PPP-dependent processes [44].

The brain slices used in our study have the advantage of preserving neuroglial interactions, but this preparation does not discriminate between the effects of PPP inhibition on glial cells or neurons. Thus, we describe the effects on all cellular compartments (processes and cell bodies) located in the CA1 region of the Stratum Radiatum, without specifying their type. Thus, our result suggests that, in the acute hippocampal slices, the PPP consumes 20% of the overall glucose and this glucose is not utilized for neuronal network activity. Previous studies on the isolated brains of monkeys [45] and rats [46] and cultured cerebellar neurons [47] assessed the contribution of glucose oxidation via the PPP to be in the range of 3–39%. This wide range, presumably, arises from using different species and experimental approaches. In general, our results are consistent with these observations.

### 3.2. Effect G6PDi−1 on on NAD(P)H/NAD(P) Levels

The oxidative branch of the PPP is known to play a significant role in the production of reduced nicotinamide adenine dinucleotide phosphate (NADPH) [22]. NADPH is a crucial cofactor involved in various metabolic processes, including the synthesis of fatty acids and cholesterol, the detoxification of reactive oxygen species, and the regeneration of antioxidant defenses [5]. Through a series of enzymatic reactions, the oxidative branch of the PPP generates NADPH by oxidizing glucose-6-phosphate and converting it to ribulose-5-phosphate [22]. This process not only provides the necessary equivalent reduction for anabolic pathways, but also maintains the cellular redox balance by ensuring an adequate supply of NADPH for various metabolic reactions.

Previous research on cultured astrocytes has shown that NADPH produced in the PPP is involved in several NADPH-dependent intracellular reactions, such as the function of cytosolic NAD(P)H-quinone oxidoreductase 1 (NQO1) [44].

In our experiments, the addition of a PPP inhibitor (G6PDi−1, 2.5 μM) did not significantly increase NAD(P)H autofluorescence and also did not alter the characteristic biphasic NAD(P)H transient induced by stimulation of the Schaffer collaterals.

The reduced cofactors (NADPH and NADH) have similar optical properties; therefore, it is expected that NADPH may contribute to the overall autofluorescence signal (approximately 18%) [48]. Apparently, for this reason, we did not observe significant changes when measuring the overall level of NAD(P)H fluorescence with modulation of the PPP by using of G6PDi−1.

### 3.3. Effect G6PDi−1 on Basic and Synaptically Induced Changes of Intracellular ROS Levels

To elucidate the effect of G6PDi−1 on basal levels and synaptically induced changes in intracellular ROS, we used the membrane-permeable fluorescent dye CellROX. Application of G6PDi−1 caused an elevation in the basal level of intracellular ROS. Low-frequency stimulation of Schaffer collaterals (10 Hz, 30 s) also caused an increase in intracellular ROS. However, this activity-induced increase in ROS was not modulated by G6PDi−1.

These observations suggest that the oxidative branch of the PPP is involved in altering the basal level of intracellular ROS, while the synaptically induced rapid increase in ROS is unaffected by this metabolic pathway.

### 3.4. Effect G6PDi−1 on Hydrogen Peroxide (H_2_O_2_) Production and Epileptiform Activity

A previous study using in vitro and in vivo models of epilepsy showed that seizure-like events are associated with the rapid release of hydrogen peroxide (H_2_O_2_) mediated by NADPH oxidase (NOX) [19]. To clarify whether the glucose utilization by the oxidative branch of PPP is involved in the regulation of H_2_O_2_ release and epileptiform activity, we analyzed the effects of G6PD inhibition in the 4AP model of epileptic seizures in hippocampal slices.

In our experiments, spontaneous seizures caused a significant transient increase in H_2_O_2_ with peak amplitudes varying from 2 to 8 μM across slices. The addition of G6PDi−1 practically did not change these values; however, it diminished the amplitude of H_2_O_2_ transients associated with interictal-like discharges and increased their frequency.

These observations might suggest the coincidence of two opposing processes: the facilitation of H_2_O_2_ release (increased frequency) due to the weakening of glutathione-dependent antioxidant defense and a decrease in NOX-mediated H_2_O_2_ production (decreased amplitude). Indeed, both processes require NADPH, the production of which is ensured by the PPP. Therefore, with its inhibition, we reduce the amount of substrate required for both antioxidant defense and oxidative stress.

The G6PD inhibitor did not affect the frequency of spontaneous seizure-like events if all data are pooled together. However, we noted that, in two out of the four slices, the drug decreased the frequency, while in the two others, it increased the SLE incidence. Such an opposite effect could be explained by the different origins of the slices used for these experiments. When cutting sagittal slices, we cannot accurately determine whether they originate from the dorsal, intermediate or ventral hippocampus. It is established that the dorsal hippocampus (DH) is involved in learning and memory associated with navigation, exploration and locomotion, whereas the ventral hippocampus (VH) is involved in motivational and emotional behavior [16]. These functions are supported by the very distinct anatomical, morphological, molecular, electrophysiological properties of hippocampal cells (see for ref. [18]). The hippocampus’ structure is also highly heterogeneous at the gene level, from its dorsal to its ventral tip [49,50]. Also, to fuel the identical neuronal network activity, DH and VH differentially recruit the energy metabolism pathways [18]. Therefore, the activity in the PPP during SLEs could be different in slices cut from the ventral or dorsal hippocampi. We tried to make transverse sections of the dorsal and ventral hippocampus using the method described in [51], but we were unable to induce spontaneous SLEs. Probably because, with these cutting methods, long-distance connections, which are necessary for the occurrence of seizures, are cut off [52].

In conclusion, this work is a first attempt to evaluate the effect of the specific G6PD inhibitor on the hippocampal neuronal network. Altogether our results suggest that G6PD inhibition suppresses the glucose consumption by the PPP. This pathway is not crucial for the generation of the normal activity of the hippocampal neuronal network; in the case of epileptiform activity, there is no clear effect in the case of the model we used to explore SLE. Further studies should be carried out using another model ex vivo and in vivo.

## 4. Materials and Methods

### 4.1. Animals

Experiments were carried out on laboratory ICR (CD-1) outbred mice of both genders, aged in different series of experiments, P20-34 and 2–3 months old. Use of animals was carried out in accordance with the Guide for the Care and Use of Laboratory Animals (NIH Publication No. 85–23, revised 1996) and European Convention for the Protection of Vertebrate Animals used for Experimental and other Scientific Purposes (Council of Europe No. 123; 1985). All animal protocols and experimental procedures were approved by the Local Ethics Committee of Kazan State Medical University (No. 10; 20 December 2016) and the AMU Ethics Committee for Animal Experimentation (#30-03102012).

### 4.2. Brain Slices Preparation

In experiments devoted to monitoring electrical activity, oxygen and glucose consummation, brain slices were prepared from 2–3-month-old male mice. The mice were anesthetized with a mixture of sevoflurane (8%) and oxygen (1 L/min) and decapitated, the brain was rapidly removed from the skull and placed into ice-cold artificial cerebrospinal fluid (ACSF). The ACSF solution consisted of (in mM): NaCl 126, KCl 3.5, NaH_2_PO_4_ × H_2_O 1.2, NaHCO_3_ 25, glucose 10, CaCl_2_ × 2H_2_O and MgCl_2_ × 6H_2_O 1.3, pH 7.3–7.4. ACSF was aerated by 95% O_2_/5% CO_2_ gas mixture. Sagittal slices (350 µm) were cut using a Leica VT 1200s tissue slicer (Leica Microsystem, Wetzlar, Germany). During cutting, slices were submerged in an ice-cold high-K^+^ concentration solution consisting of (in mM): K-gluconate 140, HEPES 10, Na-gluconate 15, EGTA 0.2, NaCl 4, pH adjusted to 7.2 by KOH. Slices were immediately transferred to a multisection, dual-side perfusion chamber with constantly circulating ACSF and allowed to recover for 2 h at room temperature (25 °C). Slices were then transferred to a recording chamber continuously perfused (9 mL/min) with ACSF (33 °C to 34 °C) containing 5 mM glucose with access to both slice sides.

For registration of ROS intracellular changes, mice (P20-34) were anesthetized with isoflurane before decapitation. The brain was rapidly removed. Sagittal 350 μm thick slice of the cerebral hemispheres containing the hippocampus were cut using a vibratome (Model NVSLM1, World Precision Instruments, Sarasota, Florida U.S.A.) in ice-cold high-K^+^ concentration solution. Slices were immediately transferred into a chamber filled with oxygenated ACSF and allowed to recover for 2 h at room temperature. Slices were then transferred to a recording chamber continuously perfused (15 mL/min) with ACSF (33 °C).

### 4.3. LFP Recording and Synaptic Stimulation

Shaffer collaterals were stimulated using the DS2A isolated stimulator (Digitimer Ltd., Hertfordshire, UK) with a bipolar electrode situated in the *stratum radiatum* of CA1 hippocampal region. Single pulses (60 to 150 µA, 200 ms) were delivered to induce a local field potential (LFP) of nearly 50–70% of maximal amplitude. LFPs were recorded using glass microelectrodes filled with ASCF, placed in CA1 *stratum oriens* and connected to the ISO DAM-8A amplifier (World Precision Instruments).

Stimulation of Schaffer collaterals (10 Hz, 30 s) was performed 3–5 times every 10–12 min in in control condition and when adding G6PDi−1 (2.5 μM) to the perfusion solution ACSF.

### 4.4. Glucose and H_2_O_2_ Measurement

Tissue glucose concentrations were measured using enzymatic microelectrodes (tip diameter 25 μm, length 0.5 mm, polarization 0.5 V, Sarissa Biomedical, Coventry, UK) connected to a TBR4100 free radical analyzer (World Precision Instruments, Hitchin, UK). Calibration was performed after the first polarization and repeated after each experiment to ensure the sensor’s unchanged sensitivity to the substrate.

### 4.5. Oxygen Measurement

A Clark oxygen microelectrode (tip diameter 10 μm; Unisense Ltd., Aarhus, Denmark) was used to measure sliced tissues’ pO_2_. The electrode was connected to a picoammeter (PA2000, Unisense Ltd., Aarhus, Denmark). A two-point calibration was performed by inserting the electrode in normal ACSF (at 33 °C) equilibrated with either 95% O_2_ or ambient air.

### 4.6. NAD(P)H Fluorescence

NADPH and NADH have similar optical properties; therefore, it is expected that NADPH may contribute to the total autofluorescence signal. Changes in NAD(P)H fluorescence in hippocampal slices were monitored using a 290–370 nm excitation filter and a 420 nm long-pass filter for the emission. Slices were epi-illuminated with monochromatic light (pE-2 illuminator, CoolLEDLtd., Andover. UK) using 365 nm. Images were acquired using a camera with a 1392 × 1040 digital spatial resolution. Due to the low level of fluorescence emission, images were acquired at 1 Hz frequency with exposition from 600 to 800 ms as 4 × 4 binned images (effective spatial resolution of 348 × 240 pixels). The exposure time was adjusted to obtain a baseline fluorescence intensity between 2000 and 3000 optical intensity levels. Fluorescence intensity changes in the stratum radiatum near the sites of LFP and O2 recording were measured in the region of interest (ROI) using the Fiji 2.9.0 software. Signal analysis was performed using the IgorPro 6.02 software (WaveMetrics, Inc., Portland, OR, USA).

### 4.7. Fluorescence Monitoring of Intracellular Reactive Oxygen Species (ROS)

CellROX Orange is cell-permeant dye that exhibits fluorescence upon oxidation by ROS. To load cells with the dye, the sagittal hippocampal slices of 20–34-day-old mice were transferred into a microchamber with 2 mL ACSF containing 5 µM of ROS-sensitive fluorescent dye CellROX Orange. Slices were incubated at room temperature with oxygenation for 40 min and then washed for 40 min with ACSF. For fluorescence monitoring, slices were placed in the recording chamber with superfused oxygenated ACSF at 33 °C. The fluorescence of CellROX Orange was excited by a wavelength of 505 nm, for 700 ms with a frequency of 1 Hz, and emission fluorescence was recorded at wavelengths above 600 nm. DriveLEDs 2.7.1. software was used to record fluorescence [53]. To monitor changes in intracellular ROS (increase in fluorescence intensity) caused by stimulation of Shaffer collaterals, regions of interest (ROI) were selected in *stratum radiatum* of the hippocampal CA1 area and in the cortex. The cortex area was used as the reference zone, where synaptic stimulation did not cause changes in fluorescence.

During the experiment, the CellROX fluorescent dye in the sections dimmed over time. In this regard, as the experiment progressed, a gradual decrease in fluorescence intensity was observed. The application of a PPP inhibitor (G6PDi−1, 2.5 μM) resulted in an increase in fluorescence levels. To assess the real change in fluorescence under the influence of G6PDi−1, the rate of change in the fluorescence level under control conditions was determined. This value was subtracted from the fluorescence level slope for the entire experimental track. This allowed us to calculate the true change in fluorescence caused by the PPP inhibitor G6PDi−1.

### 4.8. Epileptiform Activity in Mouse Hippocampal Slices Induced by 4AP

Brain slices were prepared as described in the Section 4.2 above. The prepared hippocampal slices were placed in a recording chamber with a continuously perfused ACSF solution. 4AP at a concentration of 50 μM was added to the perfusion solution, which led to the development of interictal as well as spontaneous epileptiform activity.

### 4.9. Pharmacology

The following drugs were used: G6PDi−1 (50 nM, 500 nM, 2.5 μM, 5 μM, Sigma-Aldrich, CAS No.: 2457232-14-1), CellROX™ Orange Reagent (5 μM, ThermoFisher, Waltham, MA, USA, C10443), 4-aminopyridine (4AP) (50 μM, Sigma-Aldrich, CAS No.: 504-24-5).

During recordings in brain slices, G6PDi−1 and 4AP were added to the ACSF solution with which the slices were perfused. Analysis of the effects of G6PDi−1 was carried out no earlier than 15 min after adding the inhibitor to the perfusion solution.

### 4.10. Data Analysis

Group measures were expressed as means ± SEM. Statistical significance was assessed using the Wilcoxon’s signed paired test. The level of significance was set at *p* < 0.05. Nonparametric significance tests were used to evaluate the difference between H_2_O_2_ release in control (4AP only) and after G6PDi−1 addition. We calculated the 95% confidence interval (95% CI) of the median difference. In the text, the size of the difference is presented as median difference: [lower limit, upper limit of 95% CI]. This type of analysis was performed using the estimation statistics (website www.estimationstats.com, access date: 16 December 2023). Signal analysis was performed using the IgorPro 6.02 software (WaveMetrics, Inc., USA) with custom developed macros.

## Figures and Tables

**Figure 1 ijms-25-01934-f001:**
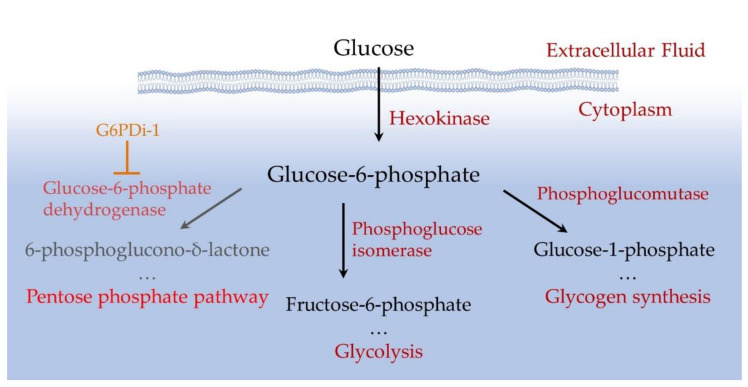
Scheme of intracellular glucose utilization.

**Figure 2 ijms-25-01934-f002:**
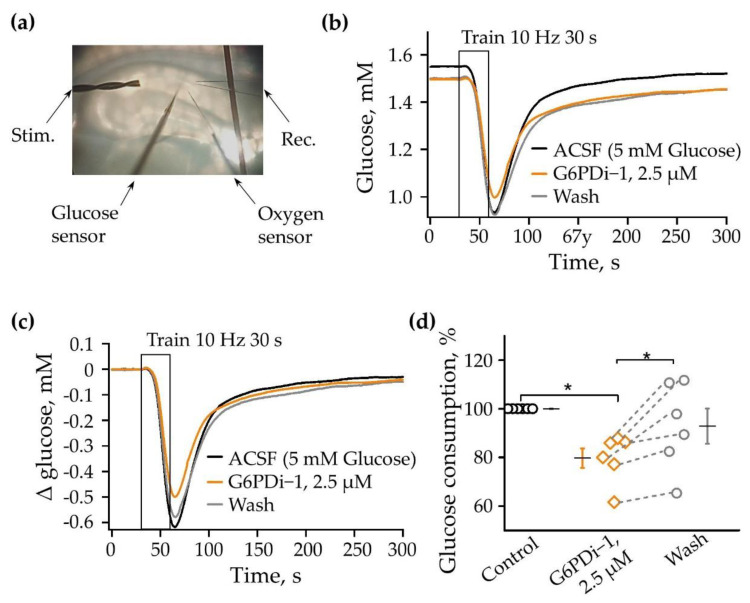
The G6PDi−1 action on glucose consumption during synaptic stimulation. (**a**) Placement of the oxygen and glucose sensors, recording and stimulating electrodes in the slices. Changes in glucose (**b**) induced by 10 Hz 30 s stimulation on one slice: in control—black line; G6PDi−1 action (2.5 μM)—orange line; and washing—gray (age of mouse: 3 months). The superimposed glucose changes are shown in figure (**c**). Summarized data showed changes in glucose, n = 6 (**d**). Each point in the group represents a separate slice, and the line shows the change in the parameter when conditions change. Black line illustrates the mean; whiskers—standard error (SE). *—*p* < 0.05 (Wilcoxson pair test).

**Figure 3 ijms-25-01934-f003:**
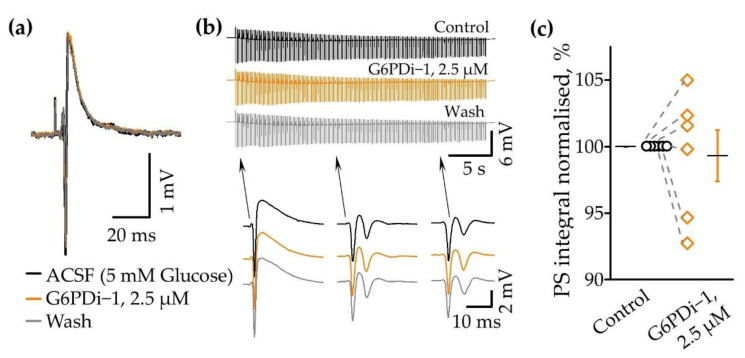
G6PDi−1 does not affect the electrophysiological characteristics of neuronal population activity. (**a**) Sample local field potential (LFP) traces from a single experiment in control (black line), during the application of 2.5 µM G6PDi−1 (orange line) and after washing (gray). (**b**) LFP traces during stimulation trains (30 s, 10 Hz) in control, in the presence of 2.5 μM G6PDi−1 and during inhibitor washout. Individual events are shown in the inserts below. (**c**) Average population spike (PS) integral values from 6 slices in control (black line), during the application of 2.5 µM G6PDi−1 (orange line) (n = 6, age 2–3 Mo). The black line illustrates the mean; whiskers—SE.

**Figure 4 ijms-25-01934-f004:**
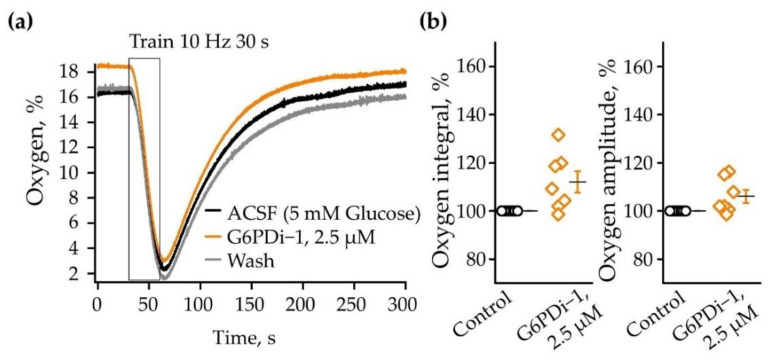
G6PDi−1’s action on oxygen during synaptic stimulation. (**a**) Changes in oxygen induced by Shaffer collaterals stimulation (10 Hz 30 s) on one slice: in control—black line; after G6PDi−1 action (2.5 μM)—orange line; and after washing—gray (Age mouse: 3 Month). (**b**) Summarized data on changes in oxygen consumption integrals and oxygen transient amplitude in control (black line), during the application of 2.5 µM G6PDi−1 (orange line) (n = 7, age 2–3 Mo). The black line illustrates the mean; whiskers—SE.

**Figure 5 ijms-25-01934-f005:**
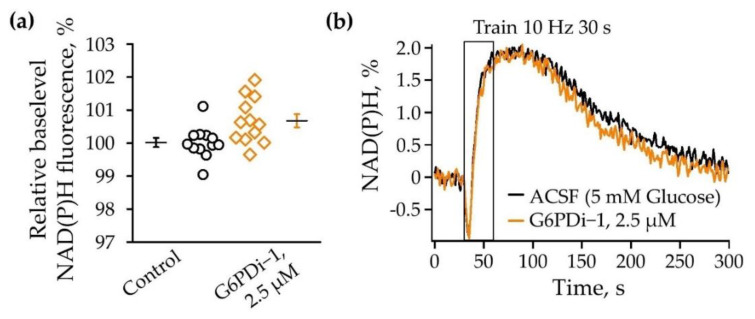
The G6PDi−1 action on NAD(P)H. (**a**) Summarized changes of NAD(P)H baselevel (n = 6, age 2–3 Mo). The black line illustrates the mean; whiskers—SE. (**b**) NAD(P)H transients induced by a 10-Hz, 30 s stimulation of Schaffer collaterals. Black signals recorded in ACSF; orange signals recorded after the addition of G6PDi−1 (2.5 μM).

**Figure 6 ijms-25-01934-f006:**
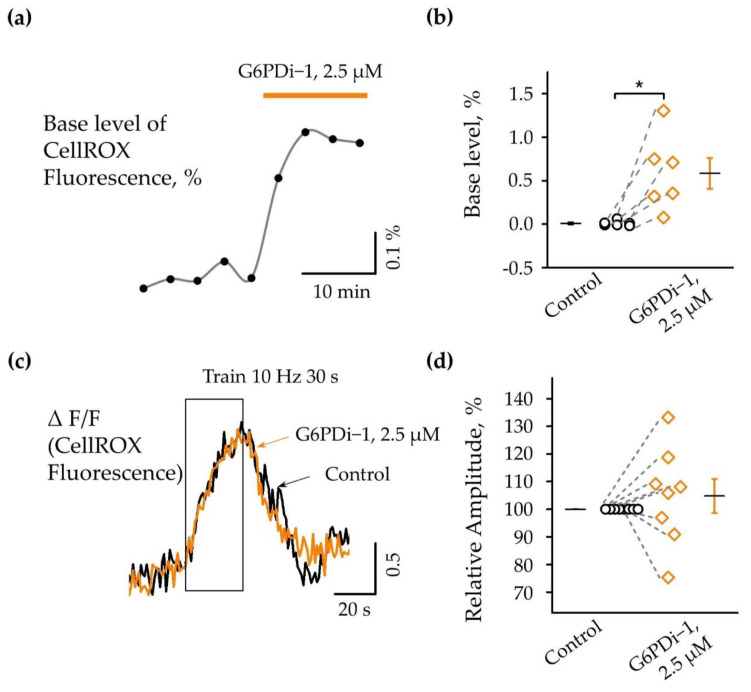
Action of G6PDi−1 on the intracellular ROS. (**a**) Base fluorescence changes in separate slices following the G6PDi−1 (2.5 μM) application, highlighted by orange line. (**b**) Summary of G6PDi−1 action on the base level of intracellular ROS (n = 6, age mice 20–25 days of postnatal development—P20-25). *—*p* < 0.05 (Wilcoxson pair test). (**c**) The graph shows superimposed accumulated signals of 2 traces of fluorescence changes during synaptic stimulation (10 Hz, 30 s) in the control (black) and during the 2.5 μM G6PDi−1 application (orange). The duration of stimulation is indicated by frame. (**d**) Summary of intracellular ROS changes under the synaptic stimulation in the control (black) and during the G6PDi−1 application (orange) (n = 8, P20-25).

**Figure 7 ijms-25-01934-f007:**
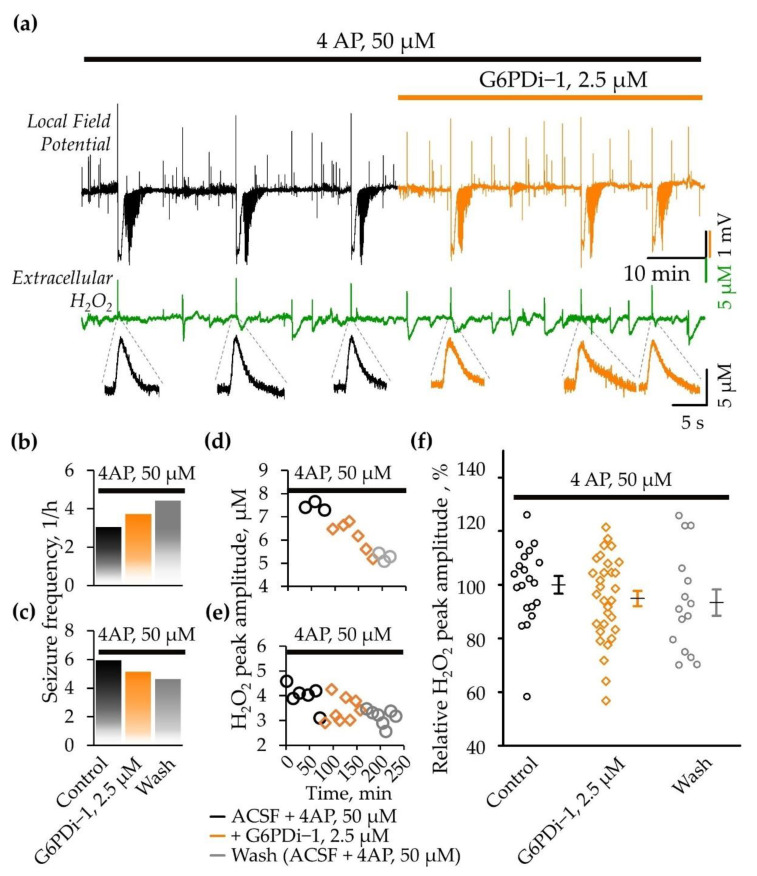
Effect G6PDi−1 on electrical activity and peak amplitude of extracellular H_2_O_2_ production during spontaneous SLEs induced by 4AP. (**a**) Representative traces of local field potential (black) and extracellular H_2_O_2_ (green) that are recorded before and after addition of G6PDi−1 (orange). Below, at expanded time scale, examples of associated extracellular H_2_O_2_ changes during spontaneous epileptiform events are shown. Frequency of spontaneous seizures (**b**,**c**) and peak amplitude of hydrogen peroxide release during spontaneous epileptiform events (**d**,**e**), that occurred in the presence of 4AP (black), after the addition of G6PDi−1 (orange) and after washing (gray), illustrated with examples from two slices. (**f**) Summary of normalized H_2_O_2_ release amplitudes during spontaneous SLEs in the presence of 4AP, after addition of G6PDi−1 and after washing (n = 4).

## Data Availability

https://disk.yandex.ru/d/jbaR4pl5-mc_Ww (accessed on 27 January 2024).

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
