# Peer review of "Analysis of the Effects of Pentose Phosphate Pathway Inhibition on the Generation of Reactive Oxygen Species and Epileptiform Activity in Hippocampal Slices"

_ijms, 2024, doi:10.3390/ijms25031934_

Round 1
Reviewer 1 Report
Comments and Suggestions for Authors
General:
In this study, the authors demonstrate, using the novel G6PD inhibitor G6PDi-1, that short-term inhibition of the Pentose-phosphate-pathway minimally affected synaptic circuit activity in mouse hippocampal slices. Furthermore, they demonstrate that G6PDi-1 influences intracellular ROS levels and synaptically induced glucose consumption but not ROS production during synaptic stimulation or in an in vitro (hippocampal slice) epilepsy model.
Overall the presented results are rather descriptive and no further insights about the underlying mechanism are presented. Additionally, no in vitro data for validation of the results are available nor information about potential translational components to human tissue. Some questions arise regarding the results and conclusions. To further strengthen the manuscript the following major and minor comments should be addressed.
Major comments:
Results:
· The authors stated that the glucose levels decrease during the inhibition of the PPP using G6PDi in a concentration-dependent manner. However, when glucose wasn’t able to be used by the G6PD, shouldn’t the concentration of the source (glucose) for G6PD not increase? Do the authors think that the unused glucose is re-channeled to other pathways (e.g.) glycolysis, therefore it might be interesting to also monitor at least the G6PD activity or Lactate (end product of glycolysis) in the experimental setup.
· Usually, the LFP is recognized using stimulation from Schaffer collaterals while the recording electrode is placed in the ipsilateral CA1 area. However, in the demonstrated picture (Fig 2a) the stimulation electrode is placed in the Schaffer collateral and the recording electrode seems to be placed in CA3. I recommend revising the picture ( or is the picture placed upside down?) and it would be helpful du indicate the different regions hippocampus within the sample.
· In Figure 3a) data for LFP “after washing” are shown, however, those data aren’t presented for the LFP during the stimulation train. Would be interesting to see if the cells react as control cells since they seem to have the same properties in the glucose experiments.
· Figure 6: Panel b/c/d/e only shows representative data, as explained in the text all tested slides seem to show inconsistent data, therefore experimental number should be increased to clarify. The authors argued that they can not ensure the same region of the hippocampus, therefore it might be possible to use different slides of one animal and compare if there are intraindividual differences.
· The measuring setups (perfusion rate, mice age, ACFS with/without? Glucose) were changed between the different experimental setups, therefore it might be questionable if the results presented for glucose and oxygen consumption can be compared with the data for H202, it would be helpful to perform additional experiments at the same setup rather for the one or other experiment.
· Figure 5c): the ROS baseline in these experiments seems to be equal in the beginning comparing WT and G6PDi traces, can the authors explain this difference in comparison to the data from Figure 5a).
Overall the authors described and potential mechanism (reduced NADPH leading to reduced NOX activity or less ROS scavenging). However, to improve the manuscript it would be helpful to perform additional experiments to clarify the potential mechanisms.
· Measurement of GSSG/GSH and/or NADH, NADPH for deeper analysis of redox state
· Analysis of NOX activity or analysis of activation of redox pathways (e.g. Nrf2)
Method section:
· ROS measurements: “In order to eliminate these effects, the trend line was determined under control conditions, and taking into account the trend, the change in fluorescence under the action of the PPP inhibitor (G6PDi-1, 2.5 μM) was calculated.„ – please clarify how the trend of ROS accumulation was taking into account?
· Please indicate if the stimulation with G6PDi1 was done in the same media (aCSF) as in the controls and indicate the Glucose concentration within the media (in Figure 2b) and 2c) it might be misleading since it looks like only the control slides hat 5mM glucose)
· Authors state a “short term” PPP inhibition, but from the method section, it isn’t clear how long the slides were incubated, and if there was a preincubation with the inhibitor prior to experiments
Discussion:
Generally, the current discussion is more repetitive and descriptive than an interpretation of the results, in this respect, it should be critically reviewed.
· The following aspects might be included:
o What do the authors think about which cell population is affected and mediated by the seen effects, is it probably to neuronal or glial metabolism?
(Watermann P, Arend C, Dringen R. G6PDi-1 is a Potent Inhibitor of G6PDH and of Pentose Phosphate pathway-dependent Metabolic Processes in Cultured Primary Astrocytes. Neurochem Res. 2023 Oct;48(10):3177-3189. doi: 10.1007/s11064-023-03964-2. )
Minor comments:
· Overall the printing quality (especially the resolution) of the figures is rather low and should be improved
· Page 2 Line 54: “Hippocampus […] emotional regulator “ – should be rephrased since mainly the amygdala and the insular cortex are involved in emotional processing
· Line 89: Citation styles switched
· Line 91: Typing error “ed”
Comments on the Quality of English Language
Reviewer 2 Report
Comments and Suggestions for Authors
Overview of the manuscript
The work focuses on analysing the involvement of G6PD dysfunction, affecting PPP, as implicated in neurological disorders, including epilepsy. The authors using the specific G6PD inhibitor G6PDi-1, evaluated its effects on mouse hippocampal slices, examining intracellular ROS, glucose/oxygen consumption, and ROS production during synaptic stimulation and in the 4AP epilepsy model.
The authors find that G6PDi-1 increased basal intracellular ROS levels but had no impact on ROS production following synaptic stimulation, furthermore G6PDi-1 didn't significantly alter spontaneous seizure frequency or H2O2 release amplitude. The authors conclude that short-term PPP inhibition has a minimal impact on synaptic circuit activity.
GENERAL COMMENT
The work is interesting and well performed. The experimental plan is constructed adequately to support the results and discussion. The methodologies adopted are valid and appropriate to analyse the topics of the work. Some points in the manuscript should be better detailed.
Specific comments
Introduction
Pag. 2, line 81: specify the acronym.
Results
Pag. 3, line 101: these concentrations of G6PDi-1 should also be reported in the Materials and Methods section
Materials and Methods
Pag. 11, line 443: the 4AP protocol should be detailed in the Mat. and Met section.
Round 2
Reviewer 1 Report
Comments and Suggestions for Authors
The authors have addressed all my comments and adapted the manuscript sufficient. However, the Figure panel about the NADPH measurements, that is described in result section is missing within the manuscript.
Furthemore there is "doubled" sentence within the discussion (double check reference as well):
Previous research on cultured astrocytes has shown that NADPH produced in the PPP is involved in several NADPH-dependent intracellular reactions, such as the function of cytosolic NAD(P)H-quinone oxidoreductase 1 (NQO1) [44]. Previous research on cultured astrocytes has shown that NADPH produced in the PPP is involved in several NADPH-dependent intracellular reactions, such as the function 386 of cytosolic NAD(P)H-quinone oxidoreductase 1 (NQO1) [48].
Author Response
We express our deep gratitude to the reviewer. Your valuable comments allowed us to eliminate errors and significantly improve the article.
We added a new Figure, removed a duplicate sentence, and corrected the citation.
Round 3
Reviewer 1 Report
Comments and Suggestions for Authors
All issues solved.